# Gastroprotective Effects of Polyphenols against Various Gastro-Intestinal Disorders: A Mini-Review with Special Focus on Clinical Evidence

**DOI:** 10.3390/molecules26072090

**Published:** 2021-04-06

**Authors:** Hui-Fang Chiu, Kamesh Venkatakrishnan, Oksana Golovinskaia, Chin-Kun Wang

**Affiliations:** 1Department of Chinese Medicine, Taichung Hospital Ministry of Health and Welfare, Taichung 40301, Taiwan; huifangchiu@yahoo.com.tw; 2School of Nutrition, Chung Shan Medical University, 110, Section. 1, Jianguo North Road, Taichung 40201, Taiwan; biochemkamesh@gmail.com; 3Department of Food Biotechnology, ITMO University, 9, Lomonosova Street, 191002 Saint Peterburg, Russia; oksana2187@mail.ru

**Keywords:** polyphenols, biotransformation, gastrointestinal diseases, gastroprotective, curcumin, resveratrol

## Abstract

Polyphenols are classified as an organic chemical with phenolic units that display an array of biological functions. However, polyphenols have very low bioavailability and stability, which make polyphenols a less bioactive compound. Many researchers have indicated that several factors might affect the efficiency and the metabolism (biotransformation) of various polyphenols, which include the gut microbiota, structure, and physical properties as well as its interactions with other dietary nutrients (macromolecules). Hence, this mini-review covers the two-way interaction between polyphenols and gut microbiota (interplay) and how polyphenols are metabolized (biotransformation) to produce various polyphenolic metabolites. Moreover, the protective effects of numerous polyphenols and their metabolites against various gastrointestinal disorders/diseases including gastritis, gastric cancer, colorectal cancer, inflammatory bowel disease (IBD) like ulcerative colitis (UC), Crohn’s disease (CD), and irritable bowel syndrome (IBS) like celiac disease (CED) are discussed. For this review, the authors chose only a few popular polyphenols (green tea polyphenol, curcumin, resveratrol, quercetin), and a discussion of their proposed mechanism underpinning the gastroprotection was elaborated with a special focus on clinical evidence. Overall, this contribution would help the general population and science community to identify a potent polyphenol with strong antioxidant, anti-inflammatory, anti-cancer, prebiotic, and immunomodulatory properties to combat various gut-related diseases or disorders (complementary therapy) along with modified lifestyle pattern and standard gastroprotective drugs. However, the data from clinical trials are much limited and hence many large-scale clinical trials should be performed (with different form/metabolites and dose) to confirm the gastroprotective activity of the above-mentioned polyphenols and their metabolites before recommendation.

## 1. Introduction

Polyphenols are classified as organic chemicals with phenolic units (three benzene rings) mostly derived from plants (phytochemicals). Polyphenols are mainly classified into five major classes including flavonoids, phenolic acids, tannins, lignans, and stilbenes. Flavonoids are again further classified into seven sub-classes including flavones, flavonols, flavanones, flavan-3-ols, isoflavones, anthocyanidins, and chalcones. Polyphenols are a secondary metabolite of plants and are normally produced on exposure to stress or pathogens (infections) and thus act as a defense mechanism to protect plants from damage [1]. Likewise, polyphenols in humans are reported to activate the antioxidant system through upregulating various endogenous antioxidants and thus effectively scavenging excessive free radicals. Additionally, polyphenols can act as a natural metal chelator and suppress the free radical generation and lipid peroxidation in humans [2]. Moreover, polyphenols are reported to show an array of biological functions other than their antioxidant activity, which includes anti-inflammation, immunomodulatory, anti-cancer, anti-diabetic as well as cardioprotective, renoprotective, neuroprotective, and gastroprotective properties [3,4]. However, the biological function of each polyphenol is related to their metabolism (biotransformation), absorption (intestinal barrier), and its utilization rate (bio-availability) as the above-indicated process is affected by various factors like structure (linkage with prosthetic groups), chemical interactions (with other macromolecules), and biotransformation process (phase I and II enzymes) [5,6]. On average, 1 g/day of dietary polyphenols are consumed and most of the bio-transformation/metabolism (90%) takes place in the large intestine. Before proceeding further, the authors would like to describe the fate of major polyphenols in humans by discussing the biotransformation process. In brief, the dietary polyphenols start to mix with the oral microbiota and undergo initial modification in the mouth with the help of various salivary enzymes and gradually pass-through the gastrointestinal (GI) tract via the esophagus and reach the stomach (acid condition), where they undergo mild enzymatic modification without any transformation. Then, the polyphenols reach the small intestine (5–10%), where the enzymatic hydrolysis of glycosides (aglycons formation) and biotransformation process (conjugation of glucuronidation, sulfation, and methylation) takes place and partially hydrolyzed polyphenols enter the liver through enterohepatic circulation and undergo further biotransformation processes (conjugation of glucuronidation, sulfation, and methylation). A portion of conjugated polyphenols enter bile and is reabsorbed back to the GI track and the remaining are effectively absorbed via systemic circulation and reach the specific cells or tissue. The remaining residual polyphenols reach the kidney and the liquid waste is excreted. Moreover, the non-absorbed polyphenols (90%), which escape from the small intestine, reach the large intestine (colon), where they undergo further enzymatic digestion by gut microbiome (bacteria) to produce various low molecular weight polyphenolic metabolites (especially phenolic acids), which are effectively absorbed (hepatic portal vein) and utilized by cells. The remaining solid waste of polyphenols is removed as feces [7,8]. The fate of dietary polyphenols (biotransformation) is shown in Figure 1.

## 2. Interplay between Polyphenols and Gut Microbiome

The human intestine, especially the large intestine (colon), contains a unique ecosystem of microorganisms (microflora), which is referred to as the gut microbiome (GM). GM is considered as part of the metabolic organ and plays a pivotal role in the maintenance of host human health status [9,10]. Studies have shown that adult healthy subjects have two major bacterial phyla including Firmicutes and Bacteroidetes. These bacterial phyla are mostly constant in adults throughout their lifetime. However, some beneficial bacteria like *Lactobacillus*, *Bifidobacterium*, *Akkermansia*, and harmful (undesirable) bacteria like *Clostridium*, *Staphylococcus*, *Enterococcus*, *Campylobacter*, and *H. pylori* might vary from person to person, based on various factors like lifestyle pattern (food pattern/diet), genes, drug, toxin/pathogens, etc. [11,12]. In particular, beneficial bacteria can produce short-chain fatty acids (SCFAs) like acetate, propionate, and butyrate, which contribute to host energy metabolism (lipogenesis, gluconeogenesis, protein synthesis), enhance the detoxification process, and thus improve overall host health status [13]. Accumulating evidence indicates that polyphenols can act as a prebiotic and enhance the production and growth of various beneficial microorganisms (probiotics) in the host gut and thereby positively modulating gut microbiota (higher beneficial bacterial and lower harmful bacterial). This, in turn, regulates the production of SCFAs, branched-chain amino acids (BCAA), vitamins, and positively modulates the metabolism of lipids and glucose, and thus promotes the overall host health status [14,15]. A recent meta-analysis conducted by Ma and Chen [16] confirmed that supplementation with polyphenol could significantly improve the production of health-promoting beneficial bacterial species like *Bifidobacterium* and *Lactobacillus* as well as suppress the production of harmful or undesirable bacterial species like *Clostridium* and *Escherichia coli (E. coli)*. Similarly, improved GM could enhance the bioavailability and bio-accessibility of various polyphenols and their metabolites during biotransformation to produce effective bioactive metabolites of polyphenols. Based on the previous evidence, it is clear that a two-way interplay exists between polyphenols and GM as they both help each other [7,12]. The overview of the interplay between dietary polyphenols and gut microbiome is illustrated in Figure 2.

## 3. Polyphenols and the Production of Various Metabolites and Their Influence on Host Health Status

Polyphenols and their bioactive metabolites are produced through the biotransformation process as above-mentioned. However, during the biotransformation process, especially in the liver, the polyphenols could result in a few adverse effects as they enhance the phase I and II biotransformation enzymes, which enhance the production of certain xenobiotics (endotoxins) and results in deleterious events like inflammation, immunomodulation, and oxidative stress [6]. Furthermore, some polyphenols can also act as a pro-oxidant and trigger oxidative stress, apoptosis, and autophagy, which might affect the normal cells. Hence, researchers have been cautious in choosing the type of polyphenol and their bioactive metabolites, especially the dosage and the mode of administration [5]. Another big disadvantage of polyphenols is the low bioavailability and in-stability, which makes polyphenol an underperforming agent (with low bio-efficiency). Hence, many scientists are working on modifying the structural and physical properties to maintain the original bioactivity of polyphenols for better biological function [17]. Table 1 lists the various bioactive metabolites and phenolic acid derivatives of popular polyphenols produced in the host during the biotransformation process. An ample number of studies have demonstrated a strong correlation between GM and the host immune system, which has paved the way for many scientists to work on polyphenols and their metabolites as they modulate GM. Thereby, they alter the host immune system via modulating various signaling transduction and improve host health status [10,13]. Additionally, polyphenols are reported to regulate the gut–brain axis and control various hormonal production as well as maintain mucus/intestinal barrier integrity in the gut, and thus dietary polyphenols and flavonoids are always considered as one of the secondary choices of preventive and therapeutic measures (complementary medicine) against numerous gastrointestinal diseases or disorders along with standard gastroprotective drugs [2,10].

## 4. Overview of the Pathophysiology of Major Gastrointestinal Disease/Disorders and Its Association with Gut Microbiome

Studies have shown that various factors including environmental factors (pollutant/smoke), dietary/lifestyle pattern (sedentary life, high fat intake-obesity, chain-smoking, antibodies/drugs abuse, excess alcohol consumption), stress, and genetical factors (some people genetic system are highly vulnerable) might contribute to dysbiosis (imbalance in gut microflora; increased harmful bacteria-pathobionts, and reduced beneficial bacteria) [18,19]. Dysbiosis results in a leaky gut (higher susceptible to pathogen intervention), which affects intestinal/epithelial integrity and thus alters immunity (mucosal barrier immunity) as well as lowers the nutrient absorption and modulate secondary bile acid (deoxycholic acid) derivative metabolism [20]. Due to the above metabolic changes, the production of SCFAs, BCAA, vitamins, and minerals are significantly reduced and also trigger the excessive production of free radicals (from secondary bile acid derivatives), eventually damaging the epithelial and mucosal cells. Hence, the gut undergoes severe stress/shock as a result of oxidative stress (imbalance in endogenous antioxidants and free radical production), which is followed by elevated inflammation (inflammasome and infiltration) in both systemic and organ-specific damage [18]. All these contributions finally end up in various gastrointestinal disorders or disease, especially gastritis, gastric cancer, colorectal cancer, inflammatory bowel disease (IBD) like ulcerative colitis (UC), Crohn’s disease (CD), and irritable bowel syndrome (IBS) like celiac disease (CED) due to a significant change in GM [15,20]. An overview of the pathophysiology of various gastrointestinal diseases and GM is well illustrated in Figure 3.

## 5. Gastroprotective Activity of Dietary Polyphenols and Their Bioactive Metabolites through Modulating GM

Based on previous reports, polyphenols and their metabolites with potent anti-inflammatory, antioxidant, anti-cancer, and immunomodulatory properties could be promising contenders in combating various gastrointestinal diseases like gastritis, gastric cancer, colorectal cancer, IBD, and IBS. Since polyphenols can positively modulate the bacterial population of the gut microbiome, they could help in regulating the host immune system, thereby improving the overall gut health [10,18,21]. Similarly, Kim and co-workers [22] indicated that polyphenols and their active metabolites enhanced the production of SCFAs and BCAA and could be useful in the treatment and prevention of various gastrointestinal disorders like Crohn’s disease, ulcerative colitis, and colorectal cancer as well as metabolic syndrome through suppressing inflammation and oxidative stress. Additionally, the authors would like to emphasize the gastroprotective property of polyphenols and hence only the mechanism related to gastrointestinal cancer, especially gastric and colorectal cancers, is discussed in this review. Since different GI diseases have different pathophysiology (as indicated before), the mechanism of treating those GI diseases with polyphenol would be in multi-directional, hence the authors compiled all of the major possible mechanisms (signaling pathway) of each polyphenol (green tea polyphenol, curcumin, resveratrol, quercetin) based on various cell lines and animal studies along with strong supporting clinical evidence.

### 5.1. Green Tea Polyphenols (GTPs)

Green tea (belonging to the *Camelia* family) is produced by a mild oxidation (fermentation) process and hence is richer in polyphenols than black tea. Green tea contains various types of polyphenols, commonly called green tea polyphenols (GTP), which include flavonoids (catechins), tannins, theaflavins, and phenolic acids [23]. Major GTPs include catechins (50–55%) like epicatechin (EC), epicatechin-3-gallate (ECG), epigallocatechin (EGC), epigallocatechin-3-gallate (EGCG), and phenolic acids like gallic acids [24]. GTP undergoes biotransformation and produces various bioactive metabolites including 4′-O-methyl-epigallocatechin (4′-MeEGC); EGCG-sulfate; EGC-sulfate, 5-(3,4′,5′-trihydroxyphenyl)-γ-valerolactone (valeric acid derivatives), and phenolic acids like gallic acid, coumaric acid, and caffeic acid (refer to Table 1 for more details) [25]. As aforementioned, catechins are the major ingredients of GTPs, of which EGCG is considered as the major catechin. EGCG possesses an array of biological activities including anti-inflammatory, anti-microbial, antioxidant, anti-platelet aggregation, anti-cancer as well as cardioprotective and neuroprotective properties [24,26,27]. Even though GTP (especially EGCG) possesses beneficial properties, its low bioavailability is one of the major issues due to its high water-solubility (unstable), and high sensitivity toward pH, temperature, and light. Hence, it is mostly combined with piperine/ fatty acids (esterified) or complexed with liposomes or nanoparticles for better absorption and assimilation [26,28].

#### 5.1.1. Proposed Gastroprotective Activity of GTP

Accumulating evidence shows that intake of GTPs rich in EGCG showed a protective effect in various human gastric epithelial and colon cell models due to its anti-invasion, anti-angiogenic, and anti-proliferative activities via modulating the Wnt/β catenin (Wingless-related integration site/β catenin) and MAPK/JNK (Mitogen-activated protein kinase/c-Jun N-terminal kinase) signaling pathways and thus effectively regulating the activation of matrix metalloproteinase (MMP-2/9) and various adhesion molecules like VCAM (vascular cell adhesion molecule) and ICAM (intracellular adhesion molecule.) [29,30,31]. Additionally, catechins of green tea are reported to show an anti-cancer effect through regulating the cell cycle (stimulate cell cycle arrest) by modulating the expression of various cyclin-dependent kinase (CDKs) and stimulating the activation of tumor suppressor genes (TSGs), thus inhibiting growth factors (vascular and epidermal). Moreover, EGCG and green tea catechins are reported to trigger apoptosis (upregulating pro-apoptotic proteins-caspase cascade) and autophagy through modulating the ERK (extracellular signal-regulated kinase), Akt, and JAK/STAT (Janus kinase/signal transducers and activators of transcription) signaling pathways in various cell and animal models [27,32,33]. The anti-carcinogenic effect (cytotoxicity) of EGCG against colon cancer has been reported due to its inhibition of DNA methyltransferases, telomerase AP-1, and histone deacetylases [34]. Supplementation of EGCG in a rat model showed a significant decrease in *H pylori*) count and in the AGS cell model, administration of GTPs and EGCG rich fraction showed better anti-*H. pylori* activity by inhibiting its adhesion and colonization via modulating the TGF-β and MMPs signaling pathway [35,36]. Furthermore, EGCG is reported to inhibit various virulent factors of *H. pylori* including Lipopolysaccharide (LPS) and cytotoxin-associated gene A (CagA) as well as inhibit urease production [37,38]. A recent study showed that EGCG can effectively inhibit *H. pylori* growth (eradicate) through binding with histone-like DNA binding protein [39]. GTP consumption showed improved *bifidobacteria*, *lactobacillus*, and *Akkermansia* bacterial (beneficial bacteria) growth, and lowered the growth of *clostridium* with improved production of SCFAs, thus maintaining gut health. Additionally, EGCG and its metabolites could inhibit celiac disease (CD) peptides (2-gliadin/32-mer peptide) absorption/metabolism [40,41,42]. Moreover, GTP could improve re-epithelization and mucous production as well as lower the incidence of inflammation/oxidative stress and thus lower the incidence of IBS and IBD [43,44].

#### 5.1.2. Clinical Evidence

Most of the clinical evidence indicates that consumption of green tea (GTP) and EGCG enriched tea or catechin-rich caffeine-free tea could significantly lower the onset and reoccurrence of gastric and colorectal cancer [27]. A clinical trial conducted by Nakachi and colleagues [45] concluded that healthy subjects, who drank 10 cups of green tea, showed a positive effect on the onset (prevention) of various cancers like colorectum, liver, and gastric. Another pilot trial also concluded that 10 cups of green tea (rich in polyphenols) consumption could substantially lower the recurrence of colorectal adenoma compared to the control group [46]. However, recent prospective studies including various trials have indicated mild or no significant impact of green tea polyphenols on colorectal cancer [47,48]. A four-week intervention with green tea polyphenols (1.2 g/day) showed a significant reduction in the number of fecal bacterial count like *clostridium* spp., with improved *Bifidobacterium* and *Lactobacillus* spp. observed in healthy subjects [49]. A small trial conducted by Boyanova and others [50] inferred that subjects (high levels of *H. pylori* infection) who drank green tea showed a significant reduction in *H. pylori* count (confirmed by urea breath test). The author pointed out that polyphenols present in green tea might help in the eradication of *H. pylori* and thus considerably lower the incidence of *H. pylori*-induced gastric ulcer and cancer. A pilot trial conducted by Dryden and his colleagues [51] observed that UC subjects treated with commercial EGCG (polyphenon E) for 56 days showed significant improvement in response rate (66.7%) and remission rate (53.3%) as well as suppressed inflammatory markers more than the placebo group. The author suggests that EGCG would be a better phytotherapeutic agent against UC along with standard immunomodulatory drugs. Overall, GTPs have shown promising results against various gastrointestinal disorders in the human model, especially against the onset and reoccurrence (relapse rate) of gastric and colorectal cancer as well as lower the adverse effect of standard chemotherapeutic agents. Additionally, its better accessibility/availability (than most of the polyphenols) and low cost have made green tea (nutraceutical/functional food) a top contender for wide use as a food supplement to lower the incidence and relapse of gastric and colorectal cancer. However, a large-scale trial with different dosage and different forms should be carried out to confirm its beneficial effects either individually or with standard gastroprotective drugs.

### 5.2. Resveratrol (Resv)

Resveratrol (3,5,4′, trihydroxy trans-stilbene) is a naturally occurring phytoalexin polyphenolic compound (stilbene class) belonging to the phytoalexin family. The major dietary source of resveratrol (Resv) includes grapes (red wine), berries, soybeans, peanuts, and pomegranates [52]. Resv exists in both cis- and trans-forms, but the trans-form of resveratrol showed better biological properties than the cis-form [53]. Resv is traditionally used to treat various ailments including pain, tissue injury, inflammatory disorders, and cancer [54]. Furthermore, it has also been reported to exhibit a broad spectrum of pharmacological activities like anti-inflammatory, anti-diabetic, anti-cancer, antioxidant, anti-convulsant, and anti-hyperlipidemic activities as well as cardioprotective, neuroprotective, and gastroprotective [55,56]. Resv is less stable with a low bioavailability and accessibility rate due to a high metabolizing rate and poor water-soluble nature. Therefore, it is mostly esterified or encapsulated, or nanosized to enhance bioavailability and accessibility [57]. The plasma half-life of free resveratrol is variable in humans and ranges from 2 to 14 h and only 0.04% of free resveratrol is excreted through urine [58]. Around 10 different resveratrol metabolites have been identified in human and rat models, and the major metabolites are resveratrol-3-O-glucuronide, resveratrol-4-O-glucuronide, resveratrol-3-sulfate, resveratrol-4-sulfate, 7,8-dihydro resveratrol-3-sulfate, 7,8-dihydro resveratrol, piceatannol, and phenolic acids like cinnamic and coumaric acid (refer to Table 1 for more details) [59].

#### 5.2.1. Proposed Gastroprotective Activity of Resveratrol

Resv is a well-known anti-cancer agent; it displays a potent anti-cancer activity in different cells/organs at different sites. Previous studies including cell line and animal models have shown that resveratrol could significantly alter various cell signaling pathways to inhibit or slow cancer cell proliferation [60,61]. During this review, the authors emphasized the gastroprotective activity of resveratrol and hence only the mechanism related to gastrointestinal cancer especially gastric and colorectal cancer is discussed here. The proposed chemotherapeutic or anti-carcinogenic effect of resveratrol against gastric and colorectal cancer are as follows: (a) resv could trigger apoptosis and autophagy through modulating MAPK/ERK, JNK, AMPK (Adenosine monophosphate-activated protein kinase), JAK/STAT, PI3K/Akt (Phosphatidylinositol-3-kinase/Protein kinase B) [60,62,63]; (b) inhibit excess epithelial cell proliferation (cell cycle arrest) by altering the proteins involved in cell cycle like cyclins and CDKs via regulating PI3K/AKT (PTEN), TGF-β, and Smad signaling pathways [64,65]; (c) suppress cell migration, invasion, and angiogenic through regulating the Wnt/β catenin and MAPK (Mitogen-activated protein kinase) signaling pathway [66]; (d) upregulate TSGs like p53 and enhance the repair system (to repair damaged DNA and mitochondria) [63]; (e) downregulate the protein expression of various pro-inflammatory cytokines by modulating the TLR (Toll-like receptor), NLRP (Nod-like receptor protein), and NF-κB (Nuclear factor Kappa B) signaling pathways [67]; and (f) improve gastric and mucous integrity by altering tight junction proteins like occluding and claudin [60]. In addition, resveratrol is reported to inhibit urease production (non-competitive inhibition) and thereby lower the pH of the gut niche, which in turn results in the death of *H. pylori* due to high acidic conditions (low survival rate of *H. pylori* in the stomach) [68,69]. Moreover, resveratrol is well known to regulate the protein expression of various virulent factors (CagA) responsible for *H pylori*-induced gastric cancer [70]. Resv also significantly suppresses the levels of IL-8 (interleukin 8) and iNOS (Inducible nitric oxide synthase) and improves the antioxidant status by upregulating the Nrf2/HO-1 (Nuclear factor-E2-related factor/Heme oxygenase-1) signaling pathway in the *H. pylori*-induced gastritis model [71]. Resveratrol administration or addition showed significant improvement in the population of *Lactobacillus* and *Bifidobacterium*, *Akkermansia* (beneficial bacteria) spp., and lowered the population of *Enterococcus* (undesirable bacteria) and thus positively regulated the host immune system, lowered inflammatory response, and thus improved the maintenance of gut homeostasis and improved overall gut health status [72]. Similarly, many researchers have confirmed that resveratrol can positively modulate the gut microflora and thus alter the immune (T cell including regulatory; Treg and T helper cells; Th) and inflammatory response (cytokines and chemokines) as well as improve antioxidant status by modulating the Nrf2/HO-1 signaling pathway, thereby maintaining gut health. Hence, resv plays a crucial role in combating IBS and IBD [73,74].

#### 5.2.2. Clinical Evidence

Few clinical trials (phase I and II) have been performed with different doses of resveratrol along with conventional anti-cancer drugs (to check synergistic activity), and the results of these few trials showed mild to moderate chemoprotective activity by suppressing the tumor growth rate (confirmed by decreased Ki-67- a proliferation marker). Some studies have reported a lower risk and onset of gastric and colorectal cancer [75,76,77]. Previous studies have demonstrated that resveratrol supplementation along with conventional chemotherapeutic drugs would significantly lower the adverse events caused by conventional chemotherapeutic drugs as well as lower the possibility of drug resistance and act as a chemosensitizer [78,79]. Meanwhile, metabolic syndrome (obese) subjects who drank red wine rich in resveratrol showed a significant increase in the count of fecal microorganisms like *Lactobacillus* and *Bifidobacteria* (beneficial microorganism) and significantly decreased levels of inflammatory markers. The results of the above trial indicated that consumption of red wine rich in resveratrol would positively modulate GM and thus favor various beneficial functions [80]. A pilot clinical trial was conducted in mild to moderate ulcerative colitis patients, who were requested to consume 500 mg of resveratrol for six weeks, showed significant reduction in inflammatory markers (tumor necrosis factor-alpha; TNF-α, high sensitivity C-reactive protein; hs-CRP, Nuclear factor-kappa B; NF-κB), clinical colitis activity index score, and increased antioxidant status with improving quality of life (QOL) [81,82]. Taken together, resveratrol could significantly improve gut function by modulating the gut microbiome and altering antioxidant, inflammatory, and immune response, which might lower the risk and onset of gastric and colorectal cancer. However, the chemotherapeutic efficacy of resveratrol depends on each person’s GM (how resveratrol is metabolized) and the stage of cancer. Therefore, large-scale trials are still needed to confirm the gastroprotective effect of resveratrol in humans. In the future, the colon-specific resveratrol delivery system might be handy for better results (gastroprotective effect) as the only fraction of resveratrol (2–5%; low bioavailability) can reach the colon and become metabolized by the gut microbiome to form a bioactive resveratrol metabolite.

### 5.3. Curcumin (Curm)

Curcumin (Curm) is one of the most popular phytocomponents of a commonly used spice called turmeric (*Curcuma longa* L). Turmeric is a sub-tropical plant that belongs to the ginger family and is used in all cuisines as a flavoring or coloring agent (yellow color) as well as for its biological functions [55]. Curcumin is one of the major curcuminoids and the main contributor to the various biological functions of turmeric. Curcumin displays a broad spectrum of biological functions like anti-inflammation, antioxidant, anti-obesity, anti-cancer, anti-diabetic, and anti-microbial as well as neuroprotective and cardioprotective properties [83,84]. The major curcumin metabolites include desmethoxycurcumin, bisdesmethoxycurcumin, curcumin-*O*-glucuronide, and curcumin-O-sulfate as well as phenolic acid derivatives like ferulic and vanillic acid [85]. The major disadvantage of curm is its limited bioavailability (poor absorption and rapid metabolism); to overcome this bioavailability challenge, curcumin is combined with piperine to form a curcumin–piperine complex or complexed with liposomes [86].

#### 5.3.1. Proposed Gastroprotective Activity of Curcumin

Curm possesses gastroprotective and healing properties through the enhancement of MMP-2 expression, along with a reduction of MMP-9 activity in gastric tissue, causing re-epithelialization and remodeling of the mucous layer (mucosal secretion and rearrangement). Curcumin can also modulate the expression of transforming growth factor (TGF β) and maintain epithelial cell integrity and microtubules/cytoskeleton [87]. Studies have demonstrated that administration of curcumin could markedly alter cancer cell division and its metastasis/invasion through modulating various signaling pathways like Wnt/β catenin, TIMPs (tissue inhibitors of metalloproteinases), and TGF β (transforming growth factor-beta) [88]. Curm can induce apoptosis or necrosis and trigger autophagy to destroy or kill infected cells by regulating the AMPK, MAPK/ERK/JNK, PI3K/Akt, and SIRT (Sirtuin) signaling pathways [89,90]. Curm supplementation in rats showed improved fecal microorganisms like lactobacillus, thus showcasing that curcumin could increase the growth of beneficial bacteria due to its prebiotic property and thus is involved in regulating the host immune system, which in turn could lower oxidative stress, inflammation, and hyper immune-activation, and eventually lower the incidence of IBD and IBS [91]. Curm treatment has been reported to eradicate *H. pylori* production and its attachment to the AGS gastric cell line due to its anti-adhesion property. Curcumin also inhibits urease activity [92,93].

#### 5.3.2. Clinical Evidence

Garcea et al. [94] conducted a trial with colorectal patients by treating them with a curcumin capsule for seven days along with conventional standard drugs and found that curcumin metabolites (curcumin sulfate and glucuronide) were present in epithelial cells of malignant colorectal calls and that they were involved in the significant reduction of leukocyte DNA adduct formation (M1G) in the colorectal compared to the placebo group. Another study also indicated that curcumin metabolites not only inhibited DNA adduct formation, but also inhibited excessive inflammation and oxidative stress induced by those adducts in epithelial cells [95]. Similarly, curcumin derivates like desmethoxycurcumin and bismethoxycurcumin are also involved in anti-inflammatory function in the animal model and UC patients [85]. A clinical trial conducted on UC patients for maintenance therapy (avoid relapse/remission) with curcumin (2 g) and conventional anti-UC drugs (sulfasalazine) for six months showed a considerable improvement in the recurrence rate (remission) and lowered the morbidity related parameters [96]. A clinical trial performed by Koosirirat and others [97] in *H. pylori* infected subjects through supplementation with turmeric capsules (enriched with curcumin, 40 mg) for four weeks showed a notable reduction in the levels of *H. pylori* and various inflammatory markers (before and after treatment with turmeric capsule). Moreover, the authors checked the integrity of the gut by evaluating fecal microorganism count, which was also improved after four weeks of intervention with curcumin. Similarly, Di Mario et al. [98] combined curcumin with two other antibodies to check whether this combination could eradicate *H. pylori*, but the results were not convincing as the treatment lasted for only one week. However, the same team tried to check the same combination of antibodies with curcumin for four weeks, and finally found a significant reduction in *H. pylori* count, which was confirmed by the urease breath test (UBT). Overall, curcumin showed some positive chemotherapeutic effects along with the standard anti-cancer drugs and was also reported to lower the prevalence of various cancers, especially gastric and colorectal cancer. However, extensive clinical trials are needed before recommending this treatment to cancer patients.

### 5.4. Quercetin (Quer)

Quercetin is a flavonol (flavonoid) that constitutes the aglycone of rutin (glycoside) and is commonly present in fruits and vegetable like onions, grapes, broccoli, apple, and berries. Quercetin glycoside and aglycon are two major forms of quercetin [99]. The quercetin glycoside (formed mainly by gut microbiome) has the best absorbable form with a longer half-life of 6–9 h, and hence has received more attention from researchers [100]. An ample amount of evidence has indicated its various biological functions, which include antioxidant, anti-inflammatory, anti-fatigue, anti-cancer, anti-microbial as well as cardioprotective, neuroprotective, gastroprotective, and hepatoprotective properties [101,102]. The major quercetin metabolites are quercetin sulfate, quercetin glucuronide, quercetin diglucuronide, quercetin glucuronide-sulfate (liver, kidney, and small intestine), and the phenolic acids (its precursor phenyl lactic acid, dihydroxy benzoic acid) and SCFAs (butyrate, acetate) are produced in the colon by gut microbiota and thus modulate host health status [103]. Similar to other polyphenols, quer is also less water-soluble with poor bioavailability and stability, therefore many researchers are trying different drug delivery systems like liposome-carriers and nanoparticles as well as esterification with other polyphenols (food-matrix) to make it more soluble and stable [104].

#### 5.4.1. Proposed Gastroprotective Activity of Quercetin

Copious cell line studies have pointed out that the administration of quercetin could notably enhance the antioxidant system through modulating the Nrf2/HO-1 signaling pathway as well as effectively attenuate the inflammatory response (COX-2 and iNOS) via regulating the NF-κB, AMPK, and TLR signaling pathways, thus decreasing the onset of various gastrointestinal cancers, particularly colorectal and gastric cancers [105,106,107]. The major mechanism by which quer exerts its chemotherapeutic activity is by inducing apoptosis and autophagy, and both of these processes are effectively activated by regulating the PI3K/Akt, MAPK/ERK, AMPK, and JAK-STAT signaling pathways [99,108,109]. Quer is also well-known to arrest the cell cycle only in cancer cells (anti-proliferative activity) by regulating CDKs and cyclin protein through various signaling pathways like PI3K/Akt, MAPK, and CREB (cAMP response element-binding) [110,111]. Anti-invasion and anti-metastasis (migration) activity of quercetin is exerted through controlling the Wnt/β catenin signaling pathway as well as its related proteins like E/N-cadherin protein (adhesion molecules) and MMPs [99,112]. It also regulates the DNA repair system, TSG production, and positively regulates tight junction proteins to ensure epithelial and mucosal integrity [113,114]. Quercetin treatment significantly lowers the inflammatory markers and alters gastric cell proliferation and apoptosis through modulating the p38 MAPK signaling pathway and thus protects the gastric mucosa from *H. pylori*-induced gastritis in the mouse model [115]. Rats administrated with quercetin showed improved gut microbiota, which in turn suppressed gut inflammation (downregulated TLR/NF-κB related proteins) and slowed or inactivated the hypertensive immune cells (Th/T17 cell), thereby decreasing dysbiosis related complications and thus maintaining gut health [116,117].

#### 5.4.2. Clinical Evidence

Numerous clinical trials are being conducted with quercetin against various metabolic syndromes such as diabetes mellitus, obesity, and some types of cancer. However, only very few clinical trials have been conducted with quercetin against gastrointestinal diseases, even though many in vitro and in vivo studies have indicated a promising gastroprotective property (maintain overall gut health) of quercetin. A clinical trial conducted with healthy subjects (with mild abnormal dysbiosis) who supplemented with quercetin showed a marked reduction in various inflammatory markers like interleukins-6/1β (IL-6/IL-1β), TNF-α, and lowered oxidative markers like lipid peroxidation products as well as increased various antioxidants. The author also hinted that a change in the gut microbiome composition (increased beneficial bacteria) might have helped in the above activity [118]. Hence, we also speculate that quercetin can be effective against various gastrointestinal diseases such as IBS and IBD. A population-based study conducted in Sweden by Ekstrom et al. [119] demonstrated a strong inverse association between quercetin and the risk of gastric adenocarcinoma due to its antioxidant, anti-inflammatory, and anti-*H. pylori* activity. A phase I clinical trial (reg at ClinicalTrial.gov-NCT00003365) conducted by treating with quercetin for 6–10 weeks in colon cancer patients showed reduced colon epithelial cell turnover markers (inhibition of colon cancer development), thus indicating the chemoprotective and gastroprotective properties of quercetin. However, the above study was not published for some unknown reason [120]. As mentioned in previous studies (animal and cell line), quercetin could be used along with conventional drugs (synergistic activity) as it acts as a chemosensitizer (improves the function of standard chemotherapeutic drugs) as well as lowers the risk of various cancers, especially colon and gastric cancer. Nevertheless, more clinical trials are needed before recommendation to combat gastrointestinal diseases, especially gastric and colorectal cancer. A brief description of the underlying gastroprotective mechanism of dietary polyphenols is presented in Figure 4.

Other minor polyphenols like cocoa flavonoids, anthocyanins (proanthocyanidins), isoflavones (daidzein and genistein), hesperidin, naringenin (citrus fruit), caffeic, coumaric, chlorogenic acid (apple), ellagic acid, and ferulic acids (phenolic acids) showed an average or mild gastroprotective activity in the clinical setting. However, none or very few clinical trials have been conducted with these polyphenols to check the gut health status [51,121,122,123]. Some studies have also indicated that the combination of various polyphenols also showed beneficial effects against gastrointestinal disorders [50,124]. In addition, few clinical trials have been conducted with the combination of specific polyphenolic compounds or their metabolites with standard chemotherapeutic drugs like tamoxifen, and sulindac for better gut health [125,126]. This mini-review has a few limitations as we included only popular polyphenols and their bioactive metabolites, which were based on strong results from the clinical trial. Additionally, during all of the clinical trials, the real effect of polyphenols could not fully measured, since it is difficult to quantify the polyphenolic content for every meal as well as the individual food materials that were consumed by the subjects every day. Hence, a well-designed clinical trial model is needed to evaluate the real-life impact of each polyphenol in humans and a proper model to check the interaction of polyphenols with every individual gut microbiome (since it has different species of the microorganism-complex system) would be useful to support the clinical trial data.

## 6. Conclusions

This mini-review summarizes the importance of various dietary polyphenols (green tea polyphenol, curcumin, resveratrol, and quercetin) and their metabolites against various gastrointestinal disorders through modulating the gut microbiome. The possible underlying mechanism was briefed with possible clinical evidence. During this review, the authors only focused on few popular polyphenols and their metabolites that possess strong antioxidant, anti-inflammatory, anti-cancer, prebiotic, and immunomodulatory activities as these polyphenols showed better results in a clinical setting than other polyphenols. Moreover, this contribution shed some light on the mechanism by which the polyphenols would lower the onset or incidence of gastric and colorectal cancer as well as mitigate the impact of other gastrointestinal diseases. Hence, the authors recommend these polyphenols along with modified lifestyle patterns and standard gastroprotective drugs for better gut health. Nevertheless, an extensive large-scale clinical trial is needed to confirm that these polyphenols can act as a therapeutic or preventive measure (complementary therapy) against gastrointestinal disease or disorders. Furthermore, each metabolite of those polyphenols should be isolated in a stable form and their beneficial properties should be tested in the future. A colon-specific polyphenol delivery system should also be developed for better chemotherapeutic efficacy.

## Figures and Tables

**Figure 1 molecules-26-02090-f001:**
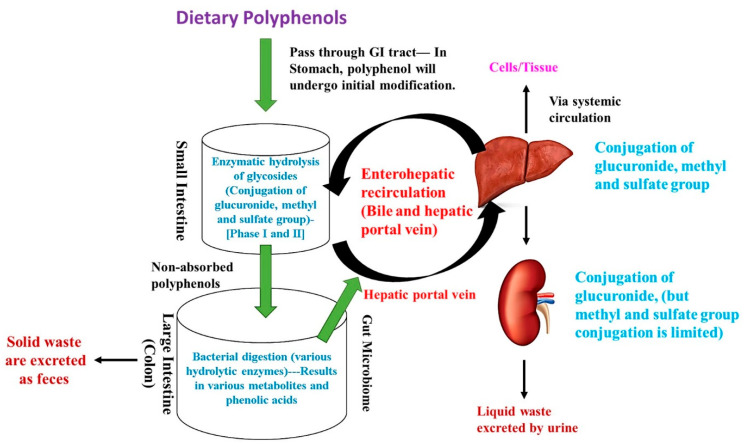
The fate of dietary polyphenols (biotransformation).

**Figure 2 molecules-26-02090-f002:**
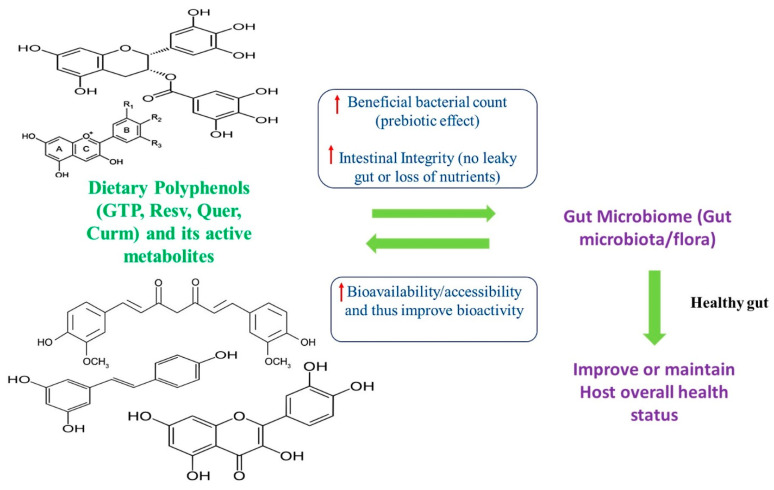
An interplay between dietary polyphenols and the gut microbiome. The upper arrow indicates an increase. GTP—Green tea polyphenol, Curm—Curcumin, Resv—Resveratrol, Quer—Quercetin.

**Figure 3 molecules-26-02090-f003:**
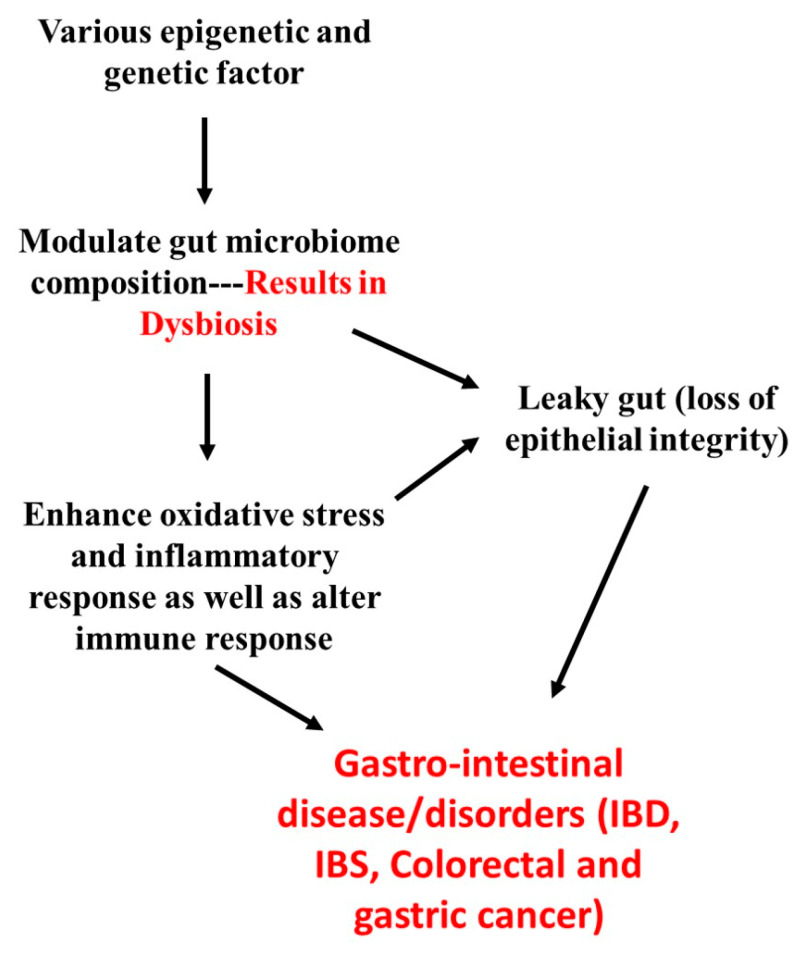
The overview of the pathophysiology of various gastrointestinal diseases and gut microbiome. IBD—inflammatory bowel disease, IBS—irritable bowel syndrome.

**Figure 4 molecules-26-02090-f004:**
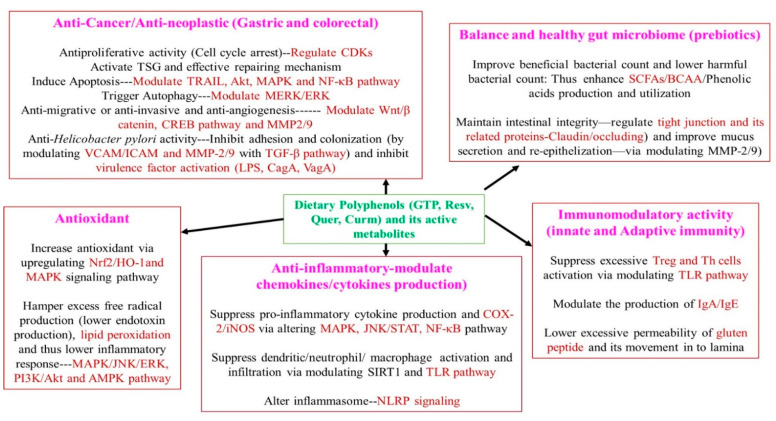
Brief description of the underlying gastroprotective mechanism of dietary polyphenols. GTP—green tea polyphenol, Curm—Curcumin, Resv—Resveratrol, Quer—Quercetin, Nrf2—Nuclear factor-E2-related factor, HO-1—Heme Oxygenase-1, NF-κB—Nuclear factor Kappa B, SIRT—Sirtuin, MAPK—Mitogen-activated protein kinase, JNK—c-Jun N-terminal Kinase, PI3K/Akt—Phosphatidylinositol-3-kinase/Protein kinase B, ERK—extracellular signal-regulated kinase, JNK—c-Jun N-terminal kinase, STAT—signal transducers and activators of transcription, JAK—Janus kinase, AMPK—Adenosine monophosphate-activated protein kinase, TLR—Toll-like receptor, NLRP—Nod-like receptor protein, CREB—cAMP response element-binding, Wnt—Wingless-related integration site, CDKs—Cyclin-dependent kinases, MMP—Matrix metalloproteinase, VCAM—Vascular cell adhesion molecule, ICAM—Intracellular adhesion molecule, BCAA—Branched-chain amino acids, SCFAs—Short-chain fatty acids, Treg—Regulatory T cells, Th—Helper T cells.

**Table 1 molecules-26-02090-t001:** List of popular polyphenols and their major metabolites.

Polyphenol	Source	Active Metabolites/Phenolic Acids and Its Derivatives
Green tea polyphenols (GTP)	Green tea leaves (*Camellia sinensis*)-Rich in Catechins	1. Methylated Metabolites: 4′-O-methyl-epigallocatechin (4′-MeEGC); 4′,4″-di-O-methyl-epigallocatechin-3-gallate (4′,4″-di MeEGCG) 2. Sulfated Metabolites: EGCG-sulfate; EGC-sulfate 3. Glucuronidase Metabolites: 5-(3,4′,5′-trihydroxyphenyl)-γ-valerolactone; 5-(3,4′-dihydroxyphenyl)-γ-valerolactone; [Valeric acid derivatives] 4. Phenolic acids: Gallic acid; coumaric acid, caffeic acid
Resveratrol (Resv)	Grape, wine, peanut, cranberry	1. Methylated Metabolites: 7′,8′-dihydro-methyl-resveratrol 2. Glucuronidase Metabolites: Resveratrol-3′-O-glucuronide, resveratrol-4′-O-glucuronide 3. Sulfated Metabolites: Resveratrol-3′-O-sulfate, resveratrol-4′-O-sulfate, 7′,8′-dihydro resveratrol-3-sulfate 4. Phenolic acids: Cinnamic acid; coumaric acid
Quercetin	Onion, apple grape, citrus fruits (glucoside form-Rutin and aglycons)	1. Methylated Metabolites: 3′-O-methyl-quercetin, 4′-O-methyl-quercetin 2. Glucuronidase Metabolites: Quercetin-3′-O-glucuronide, Quercetin-4′-O-glucuronide, Quercetin-3′-4′-di-O-glucuronide 3. Sulfated Metabolites: Quercetin-3′-O-sulfate Lots of lactic and benzoic acid derivatives like dihydro-phenylacetic acid/propionic acid/benzoic acid).
Curcumin	Turmeric (*Curcuma longa*)(Curcuminoid)	1. Methylated Metabolites: Desmethoxycurcumin; bisdesmethoxycurcumin 2. Glucuronidase Metabolites: Curcumin-O-glucuronide, Di/tetra/hexa/octa-hydro-curcumin glucuronide 3. Sulfated Metabolites: Curcumin-O-sulfate, Di/tetra/hexa/octa-hydro-curcumin sulfate 4. Phenolic acid: Ferulic and vanillic acid as well as dimethyl form.

## Data Availability

Not applicable.

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
