# Peer review of "Gastroprotective Effects of Polyphenols against Various Gastro-Intestinal Disorders: A Mini-Review with Special Focus on Clinical Evidence"

_molecules, 2021, doi:10.3390/molecules26072090_

Round 1
Reviewer 1 Report
The authors' work on the review “Gastroprotective effects of polyphenols against various gastrointestinal disorders: A mini-review with special focus on clinical evidence” is exiting and indeed of importance to the public and the wider scientific community.
However, there is poor grammar coupled with far too long sentences throughout the manuscript which makes the content very difficult to read. As such the manuscript cannot be published as they are. The authors need to comprehensively review the manuscript to ensure easy readability.
Below are examples of poor sentence and grammar structures and this is only in the abstract.
Line 16 -21 – separate sentence in two otherwise it’s too long.
Line 21 - change author to authors
Line 22 – reword sentence – doesn’t make sense otherwise
Line 25 – change “polyphenol” to “polyphenols”
Line 29 – Change “those above-mentioned to “the above mentioned”
Line 36 change “majorly” to “mainly”
Line 81 – 86 – The authors have implied that some bacteria may vary, there are substantial data to show that the firmicutes and Bacteroidetes ratios can change with diseases status – Could the authors elaborate on this further? i.e whether changes are observed across phylum, genera, species and so on.
Line 139- Authors have used the word “epigenetic to refer to various factors influencing the microbiome” this is an incorrect use of the word epigenetic as it usually refers to “heritable phenotype changes that do not involve alterations in the DNA sequence examples being DNA methylation and histone modification, each of which alters how genes are expressed without altering the underlying DNA sequence. Please use a different word
I suggest that the manuscript be accepted after extensive editing of English language otherwise it is not easy to read.
Author Response
Comments and Suggestions for Authors by reviewer 1:
The authors' work on the review “Gastroprotective effects of polyphenols against various gastrointestinal disorders: A mini-review with special focus on clinical evidence” is exiting and indeed of importance to the public and the wider scientific community.
However, there is poor grammar coupled with far too long sentences throughout the manuscript which makes the content very difficult to read. As such the manuscript cannot be published as they are. The authors need to comprehensively review the manuscript to ensure easy readability.
Thanks for the comments, we recast few sentences for better understanding and easy readability (highlighted in yellow).
Below are examples of poor sentence and grammar structures and this is only in the abstract.
Line 16 -21 – separate sentence in two otherwise it’s too long.
Modified as indicated
Line 21 - change author to authors
Modified as indicated
Line 22 – reword sentence – doesn’t make sense otherwise
Recasted as indicated
Line 25 – change “polyphenol” to “polyphenols”
Modified as indicated
Line 29 – Change “those above-mentioned to “the above mentioned”
Modified as indicated
Line 36 change “majorly” to “mainly”
Modified as indicated
Line 81 – 86 – The authors have implied that some bacteria may vary, there are substantial data to show that the firmicutes and Bacteroidetes ratios can change with diseases status – Could the authors elaborate on this further? i.e whether changes are observed across phylum, genera, species and so on.
Many studies have found that consumption of polyphenols would alter the composition of gut microbiome especially improve the beneficial bacterial species counts (Lactobacillus, Bifidobacterium, Akkermansia spp) and lower the undesirable or harmful bacterial counts (Clostridium, Staphylococcus, Enterococcus, Campylobacter, Helicobacter spp). Thereby improve host immunity and thus enhance host overall health status. However, the changes by polyphenols in the Bacteroidetes and firmicutes counts are observed across different species. Nevertheless, no specific information was revealed to showcase that particular genus or phylum microbes could contributes for beneficial or undesirable effects. Since, different polyphenols are metabolized different by different gut microbiota. Hence, we brief those details in our manuscript body.
Line 139- Authors have used the word “epigenetic to refer to various factors influencing the microbiome” this is an incorrect use of the word epigenetic as it usually refers to “heritable phenotype changes that do not involve alterations in the DNA sequence examples being DNA methylation and histone modification, each of which alters how genes are expressed without altering the underlying DNA sequence. Please use a different word
Sorry for the usage of wrong word. We modified it accordingly.
I suggest that the manuscript be accepted after extensive editing of English language otherwise it is not easy to read.
Also, the whole manuscript was cross-checked for grammatical errors by native English speakers and all the changes were highlighted in yellow.
We appreciate the reviewers for providing positive feedbacks/ comments to recast our review paper.
Reviewer 2 Report
Comments to the Authors:
The paper focuses on “Gastroprotective effects of polyphenols against various gastrointestinal disorders.”
Overall, the compilation was done reasonably. The manuscript has so many grammatical errors and spelling mistakes. I would recommend the authors to choose English editing services from MDPI to fix these errors.
The authors should use the correct abbreviations throughout the manuscript including figures. E.g., Figure 2: Curcumin, Resveratrol, Quercetin were abbreviated as Curm, Resv, Quer, respectively. These compounds were never abbreviated the same way in the text elsewhere in the manuscript.
I would be happier if the authors can include a few more compounds to make this manuscript comprehensive. At present, the authors have compiled only a few compounds in this review.
What is the rationale behind choosing these polyphenols, and what do you mean by “popular polyphenols” (line 489)?
Only one table in the whole review. Table 1 is the list of compounds. There has to be at least one more table where data can be represented in tabular form.
Author Response
Comments to the Authors by reviewer 2:
The paper focuses on “Gastroprotective effects of polyphenols against various gastrointestinal disorders.”
Overall, the compilation was done reasonably. The manuscript has so many grammatical errors and spelling mistakes. I would recommend the authors to choose English editing services from MDPI to fix these errors.
The whole manuscript has been revised and few grammatical and typo graphical errors are resolved in the revised manuscript.
The authors should use the correct abbreviations throughout the manuscript including figures. E.g., Figure 2: Curcumin, Resveratrol, Quercetin were abbreviated as Curm, Resv, Quer, respectively. These compounds were never abbreviated the same way in the text elsewhere in the manuscript.
We also used those abbreviation in our manuscript, but limited. However, for each figure caption we will give an abbreviation as well so that the reader can easily understand.
I would be happier if the authors can include a few more compounds to make this manuscript comprehensive. At present, the authors have compiled only a few compounds in this review.
We also love to include, but our focus on only few popular polyphenols (nutraceuticals/ functional foods) with promising results in humans. We clearly mentioned in the limitation section that only those polyphenols (green tea polyphenol, curcumin, resveratrol, quercetin) showed good or positive impact on human gut health. Moreover, this is a mini-review and thus we discussed only few polyphenols. Thanks for the comments, sure we will come up with new comprehensive review including all the polyphenols and its importance in gut health especially in human.
What is the rationale behind choosing these polyphenols, and what do you mean by “popular polyphenols” (line 489)?
As we indicated before, this mini-review focus only on few popular polyphenols as they showed positive impact on human gut health. Which are well supported by including many clinical trials for each polyphenol. Also, all the polyphenols which were discussed in this review should be commonly available throughout the world (easily available or used) as well as those polyphenols should also comes under nutraceuticals or functional food categories so that these polyphenols can be easily used as a supplements or complementary therapy.
Only one table in the whole review. Table 1 is the list of compounds. There has to be at least one more table where data can be represented in tabular form.
Yes, we have only one table, but we have included 4 figures, which will elaborate all the details especially the figure 4, which will compile all the gastroprotective mechanism behind polyphenols.
All the changes were included in our revised manuscript as indicated by the reviewer.
Also, the whole manuscript was cross-checked for grammatical errors by native English speakers and all the changes were highlighted in yellow.
We are grateful for the reviewers to giving a positive comments or suggestions.